# Study of Ageing in Complex Interface Interaction Tasks: Based on Combined Eye-Movement and HRV Bioinformatic Feedback

**DOI:** 10.3390/ijerph192416937

**Published:** 2022-12-16

**Authors:** Ting Huang, Chengmin Zhou, Xin Luo, Jake Kaner

**Affiliations:** 1College of Furnishings and Industrial Design, Nanjing Forestry University, Nanjing 210037, China; 2Jiangsu Co-Innovation Center of Efficient Processing and Utilization of Forest Resources, Nanjing 210037, China; 3School of Art and Design, Nottingham Trent University, Nottingham NG1 4FQ, UK

**Keywords:** complex interaction tasks, age-friendly interface, eye tracking, multi-physiological signals

## Abstract

Human–computer interaction tends to be intelligent and driven by technological innovation. However, there is a digital divide caused by usage barriers for older users when interacting with complex tasks. To better help elderly users efficiently complete complex interactions, a smart home’s operating system’s interface is used as an example to explore the usage characteristics of elderly users of different genders. This study uses multi-signal physiological acquisition as a criterion. The results of the study showed that: (1) Older users are more attracted to iconic information than textual information. (2) When searching for complex tasks, female users are more likely to browse the whole page before locating the job. (3) Female users are more likely to browse from top to bottom when searching for complex tasks. (4) Female users are more likely to concentrate when performing complex tasks than male users. (5) Males are more likely to be nervous than females when performing complex tasks.

## 1. Introduction

The development of intelligent technology and changes in information technology, on the one hand, tend to make some activities more engaging and more efficient. On the other hand, new technology is not intuitive to older persons, who are the most socially disadvantaged group. The digital divide is further distancing elderly users from modern society, leading to inconveniences and difficulties in their lives. In the post-epidemic era, mobile health products, such as those involving intelligent interface interactions, are difficult to use for older users and are inconvenient for travel [1,2,3,4]. The development of intelligent homes brings not only the upgrading of hardware systems but also the construction of software systems [5,6]. Most of the research on age-appropriate interfaces for smart homes continues with the approach of traditional intelligent interfaces. Taking intelligent home systems as another example, the market’s current mainstream class of intelligent home system operations is often equipped with control energy applets or product applications to complete intelligent interactions. However, the application’s interface tends to be complex for completing complex interaction tasks with the intelligent home system. The high density of information in the interface also leads to a low completion rate and low response rate of the actual operation.

In intelligent interface research, eye-tracking technology is regarded as a reliable approach [7,8,9], including the development of head-mounted eye-tracking technology and virtual eye-tracking technology [10,11,12]. Most research on intelligent interfaces has focused on design elements: layout, swipe patterns, icon size, etc. [13,14,15,16]. Research has shown that older users do not have the same concerns as younger people when using intelligent interfaces. Older users prefer interaction on small screens such as smartphones to human–computer interaction on large screens [17,18,19]. Older users of different genders also focus on different response points when completing interface interactions. A flat design tends to be popular in interface styles [20]. In line spacing studies, multiple experiments have shown that wide line spacing and large fonts effectively improve reading accuracy and reduce reading fatigue [21,22]. In terms of swiping, horizontal swiping is more conducive to older users completing tasks [23]. Older users preferred mimetic icons in terms of icon design [24,25]. During two or more independent task interactions, older users’ completion performance was significantly lower than that of younger users, indicating barriers to transmitting information [26,27]. The comprehension and perceptions of older users were low for menu grouping, but research has shown that interaction was not best using a single test [28,29,30,31]. In intelligent home service interaction systems, operational tasks are complex; they include increased interaction steps and high task failure rates for older users [32,33,34,35]. At the same time, the intervention of complex colours can also affect the efficiency of older users [36,37,38].

This study, therefore, focuses on the optimal menu grouping of older users of different genders when performing complex tasks and the changes in physiological signals when completing performance. Based on the self-designed intelligent home interface operating system, the concerns of elderly users of different genders while completing complex tasks are explored. Based on this, suggestions are provided for an age-appropriate interface for smart homes and a reference for precisely tailored products.

## 2. Methodology

### 2.1. Participants

Twenty-one males and twenty-one females from 60 to 75, all right-handed, with normal or corrected-normal vision, were chosen for this study, as shown in Table 1. It is worth stating that all subjects were required to be free of eye disease and have good eyesight. For ethical compliance, all subjects who participated in the research signed an informed consent form, and all experiments were carried out with the subjects’ consent.

To establish if the senior subjects were familiar with mobile phone applets, a combination of subjective questionnaire measurement and on-site observation were used. The subjects were chosen In part based on how frequently (daily, weekly, monthly, etc.) they used a smartphone-related application (20% of the score). Then, 80% of the score came from watching them complete tasks that included step-by-step instructions on how to use the application, and looking at various levels and types of detailed descriptions to assist the subjects in completing the tasks on their own without the observer being involved in the completion of the tasks. The Ordinary/Observational Score (OS) score was used to calculate the OS scores for the subjects (OS = 20% frequency of usage + 80% job completion and time spent). The OS score ranges from 0 to 10, with 10 being the highest. Users’ familiarity with mobile applications is categorised into three groups based on a cutoff score of 5. Two groups were formed from the "high-level group" and the "low-level group." The familiarity levels of the 39 subjects were balanced by having 8 with high familiarity and 11 with low familiarity.

### 2.2. Experimental Sample Design

An applet for a self-created home control system was installed on a smartphone, which was used for the experiment. User interactions and the product interface were both photographed using two digital cameras, and test data were saved for later processing. Smartphone: full screen, 6.22 in., FHD+1, 520 720 px. Figure 1 depicts the prototype user interface.

### 2.3. Experimental Procedure

The elderly subjects were told to operate the intelligent home software system using an applet to operate and adjust the angle of the intelligent sofa, as shown in Figure 2. The experimental equipment involved multi-channel physiological monitoring: ErgoLAB Biosensing and Eye movement meter; Tobii Pro Glasses 2. The physiological monitoring was used to observe the emotional changes of the subjects during the experimental operation to help them complete the instructions and to simplify the operational behaviour. The oculomotor was used to observe the subject’s eye movements, attention points, points of interest, etc., during behavioural interactions.

(1)The experimenter in the field wore a wireless physiological monitoring device, which was used to monitor the HRV (Heart Rate Variablity) and EDA (Electrodermal Activity) data of the elderly subjects to determine their stress level during the task.(2)The geriatric subject was fitted with Tobii Pro Glasses 2. Calibration of the oculomotor was performed prior to the start of the experiment. The subject was instructed to place the calibration card at arm’s length, and the eye-tracking technique was used to calibrate the eye-tracking device. During the experiment, the subject operated the experimental handpiece installed with the program to adjust the experimental seat autonomously. During the mobile phone’s operation, the subject must remain approximately 40 cm from the mobile phone’s screen.(3)The elderly subject needed to sit in the experimental chair, find the button for seat adjustment on interface one (as shown in Figure 3a), find the corresponding icon, and click on the adjustment slider.(4)After the adjustment slider was invoked, the elderly subject needed to adjust the second slider bar (shown in Figure 3b) and drag it to 125°.(5)At the end of the task, the field testers removed the experimental equipment for the elderly subject.

**Figure 3 ijerph-19-16937-f003:**
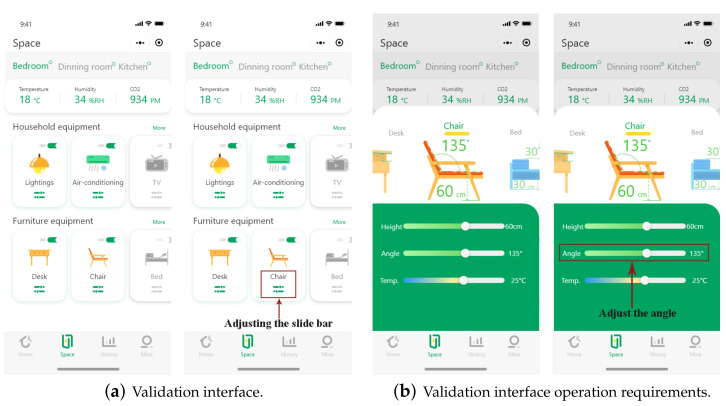
Interface for verification.

## 3. Statistical Results and Analysis

### 3.1. Eye Movement Data Collection

The average pupil diameter of the subjects increased during the first part of the experiment, peaked, and then began to decrease. The subjects were exposed to the home control system applet based on the dynamic experimental procedure. Therefore, their pupil diameters were in two dynamic phases, the first being the rising phase, which can be referred to as the stimulus arousal zone. Subsequently, as the stimulus continued, the subject began to feel fatigued, and the novelty decreased, leading to the fatigue phase. Astigmatism during eye movement data acquisition also affects calibration. This limits the range of user groups for eye interaction, but phakic or myopic eyes have little effect on the acquisition of eye movement data. A total of 208,200 pupil diameter data were collected for a single person, and 8,744,400 EDA (Electrodermal Activity) data were collected for a single subject. The mean values of pupil diameter change for the different stages were classified according to the behavioural stage.

Campenhausen found that individuals who focused on a single stimulus experienced a significant increase in eye velocity and a decrease in look nystagmus [39]. Studies have shown that eye movement indices such as peak sweep speed and blink frequency are related to the attentional control process [40]. To further investigate the specific grading of this process, Bachurina experimentally investigated the relationships between eye movements and mental attention tasks, concluding that peak sweep speed and blink frequency decreased with mental attention demands and were negatively correlated with self-evaluation of mental work [41]. Eye movement velocity is also used in practical occupational safety assessments to assess the attentional focus of actual workers. Zhu used video capturing of workers working with safe and unsafe behaviours to determine the level of attention by eye movements [42]. It was found that workers’ attention levels were higher during high-risk operations than in safe operations and that workers’ eye movement speed was higher in risky than in safe operations. The eye movement data focus on pupil diameter, blink count, etc. We counted the data of each type of eye movement of the subjects of different genders while performing the interface operations, as shown in Table 2 and Table 3. Table 2 represents the eye movement data for subjects of different genders during manipulation task 1, and Table 3 represents eye movement data for subjects of different genders during manipulation task 2. The eye movement data under each task module was subdivided into data for the left and right eyes, comprising four sets of items—1: eye-velocity processing data, 2: left-pupil processing data, 3: right-pupil processing data, and 4: mean-pupil processing data. The data were mainly analysed for maximum, minimum, sd, variance, mean and median. During the overall operation of interface 1, the primary interaction task was to find the intelligent sofa operation panel, which can be categorised as a complex task of finding and locating.

The main interaction task in Task 1 was to find the adjustment icon for the intelligent sofa, which can be considered a finding task in a complex panel. For the eye-velocity processing data during this task, it can be seen in Table 2 that the average time taken for the male elderly subjects to complete this task was 25.6 s, whereas it was 15.83 for the female elderly subjects. Multiple statistics suggest that males maintained high levels of concentration for more extended periods than females when completing the finding task. The main task in Task 2 was adjusting the tilt angle of the intelligent sofa, which can be categorised as a complex task of goal attainment. For the eye-velocity processing data of this task, it can be seen from Table 3 that the average of male elderly subjects when completing this task was 11.46 s, whereas it was 17.04 s for female elderly subjects. The multiple data values suggest that females were more attentive than males in completing the task.

Blink behaviour was often used to indicate internal and external attentional focus. Blinking interrupts visual input, temporarily inhibits visual processing, and is reduced when the task requires processing external visual information [43,44,45]. Wide eyes are associated with experimental tasks of this type, whereas browsing tasks are external tasks requiring frequent sweeping of one’s surroundings [46,47]. Annerer, in order to explore the relationship between wide eyes and attention in different task situations, used experiments to simulate different task types and verified through machine learning that in visuospatial tasks, the external focus of attention was associated with more wide eyes than internal attentional focus [48]. Furthermore, the sweeping gaze remained consistently high in high-attentional situations. This experiment recorded the number of blinks, the average number of blinks, the number of saccades, the average number of saccades, and the time taken for saccades for all subjects while performing complex external interface operations, as shown in Table 4. The average number of saccades for male elderly subjects was 3.37, whereas for female elderly subjects, it was 3.42. Therefore, this indicates that female elderly subjects were more attentive than male elderly subjects throughout the experiment.

The pupil diameter reflects the subject’s mental load: a larger pupil diameter reflects a more significant mental load. The pupil diameters of the different genders were measured and counted. Table 5 shows that the average pupil diameter of male subjects during the complex interaction task was 4.26 mm, which was larger than that of female subjects. This directly reflects that the mental load of the male elderly subjects was higher than that of the female elderly subjects during the interaction with the intelligent home software interface.

Heatmaps can be used to discover the visual objects that attract the most attention, compare the strengths and weaknesses of the visual objects concerning the user’s attention, and have the advantage of supporting multi-user data displays. Figure 4a–d show the hotspot and eye-tracking graphs for men during page browsing and task completion, and Figure 4e–h show the hotspot and eye-tracking graphs for women during page browsing and task completion. In the search task for interface 1, the hotspot map shows that female subjects were more likely to be attracted to the rest of the interface, combined with the trajectory map. Female subjects continued to browse the rest of the interface after searching for the right panel. They were also more attracted to the icons than the textual information, and their eyes stayed longer. In the manipulation task of interface 2, the hotspot diagram reflects that the female’s attention is entirely focused on manipulating the slider bar. In contrast, the male subjects’ eyes are also drawn to the rest of the iconic information.

### 3.2. HRV Data Collection

The HRV physiological signal acquisition of elderly subjects was carried out. The HRV physiological signal can be divided into an ECG (Electrocardiogram) signal and a PPG (PhotoPlethysmoGraphy) signal. The HRV signal was first pre-processed using the wavelet noise reduction method. The frequency range of the ECG signal was 0.01∼200 Hz; the frequency range of the pulse signal was 0.1∼40 Hz. The noise signal was removed by high-pass and low-pass filtering, and the influential data band was retained. Band-stop filtering is switched on mainly to remove industrial frequency interference from the environment by locating intervals that vary from the previous interval by more than a user-defined percentage (typically 20%) as irregular intervals. The anomaly detection formula is shown in Equation (Equation 1).
(1)$$\begin{array}{c} D\left(n\right)=\frac{\left|x\left(n\right)-med\left(x\right)\right|}{1.483\times med\left|x\left(n\right)-med\left(x\right)\right|} \end{array}$$

The mean-square correction was used for ectopic intervals. The mean-square method replaces the ectopic interval with the mean of the adjacent IBI intervals centred on the ectopic interval, as shown in Equation (Equation 2). Similarly, the median method replaces the ectopic interval with the median of the adjacent IBI intervals. Finally, it replaces the ectopic interval with the cubic spline function, as shown in Equation (Equation 3).
(2)$$\begin{array}{c} {\mathrm{ibi}}^{\prime }\left(n\right)=\mathrm{mean}\left(\mathrm{ibi}\left(n+m\right)\right),\mathrm{where}\;\left|m\right|⩽\frac{\left(w-1\right)}{2} \end{array}$$
(3)$$\begin{array}{c} {\mathrm{ibi}}^{\prime }\left(n\right)=\mathrm{med}\left(\mathrm{ibi}\left(n+m\right)\right),\mathrm{where}\;\left|m\right|⩽\frac{\left(w-1\right)}{2} \end{array}$$

Time-domain HRV analysis evaluates heart-rate variability by calculating a series of mathematical and statistical indicators of the R-R interval, revealing the pattern of signal changes over time. The time-domain indicators mainly correspond to the magnitude of sympathetic and parasympathetic tone and thus to the overall degree of activation of the autonomic nervous system. Our ECG signals had a short duration, so they were mainly processed in the RMSSD band for time-domain analysis, as shown in Equation (Equation 4).
(4)$$RMSSD=\sqrt{\frac{1}{N-1}\sum_{i=1}^{N-1}(RR_{\left(i+1\right)}-RR_{i})2}$$

The RMSSD bands were processed for time-domain analysis, as shown in Table 6. The combined statistical analysis in Figure 5 shows that the RMSSD bands of the elderly female subjects were much larger than those of the males. This reflects that their heart rate is lower than that of males when performing complex tasks, and they are in a more relaxed state.

## 4. Conclusions

In experiments, it was observed that older users often experience negative emotions, such as nervousness, when interacting with complex tasks. Therefore, the interface should be simple and display only minor task information and interface elements. In this experiment, real-time data collected from multiple physiological signals (eye movement data, HRV data, EDA data) were used to make the following conclusions: (1) Older users look at icons first before focusing on text parts during human–computer interaction. (2) When searching for complex tasks, female subjects first navigate through the global page before looking for basic tasks. (3) Females’ reading habits are top-down. Male subjects were used to looking directly for the critical task rather than focusing excessively on the global interface. (4) Males were more focused than females when performing a finding task with a complex interface and remained focused throughout. (5) During interaction with complex tasks, female subjects experienced more minor mood swings and tension than male subjects.

Based on the above studies, simple icons are preferred over complex verbal annotations when adapting to complex interface designs. In complex task interactions, the operational steps will be further simplified to reduce the operational difficulty and eliminate the uneasiness of older users. When personalising for male and female older users, the elderly-male user interface should be more task-focused, with a simplified and eye-catching command panel. When personalising the user interface for female users, it should be oriented towards operational tasks, with the Find Task module being visible and easy to find. This will optimise the experience of older users and improve their performance.

There are still things that could be improved in this experiment. (1) The subjects operated the mobile phone independently during the experiment. To a certain extent, their ability to operate electronic devices may have impacted the results. (2) The distance between the subject’s eyes and the screen was controlled: 40 cm. However, the subject’s habits of using electronic products, such as leaning forward and leaning back, could be studied in depth based on the pupil’s vertical distance and other data. (3) This experiment lacked personal-scale collection for the subjects, and the subsequent experiment will combine subjective and objective data to establish a multi-dimensional assessment system. (4) This experiment combined three channels of eye movement, HRV, and EDA signals to investigate the behavioural characteristics of elderly male and female subjects of different genders when performing complex tasks. A follow-up experiment will further integrate EEG (Electroencephalogram) and other psychological load indicators to investigate older subjects’ performance characteristics in complex tasks.

## Figures and Tables

**Figure 1 ijerph-19-16937-f001:**
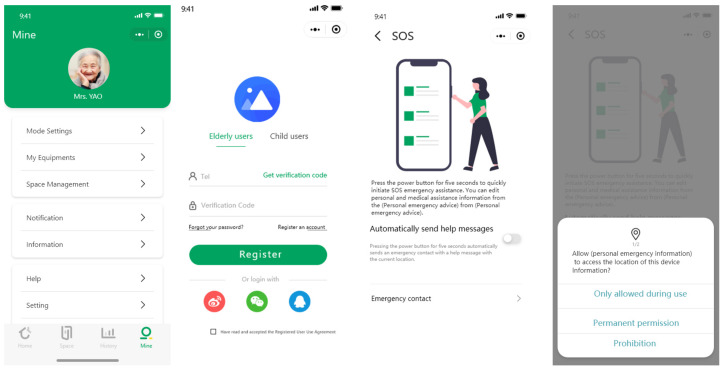
Elderly smart phone use. Designed in-house by the team, the overall colour palette was chosen in accordance with the principles of appropriate ageing design.

**Figure 2 ijerph-19-16937-f002:**
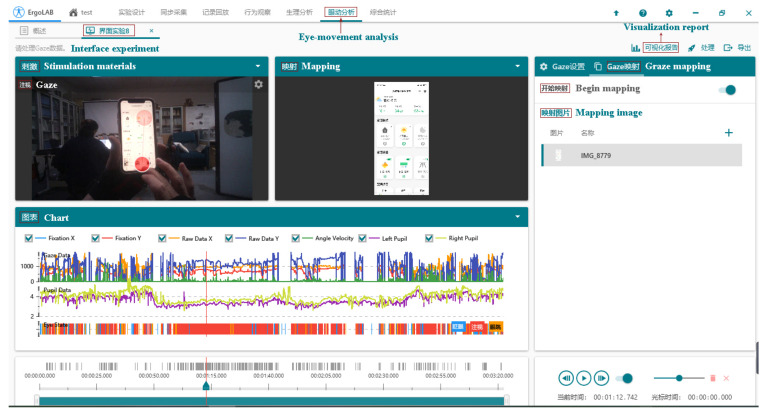
The platform for experimental field runs. It consists of three main windows: live video and eye movement data filmed real-time, real-time mapping of stimulus materials, and real-time monitoring of physiological sensor data.

**Figure 4 ijerph-19-16937-f004:**
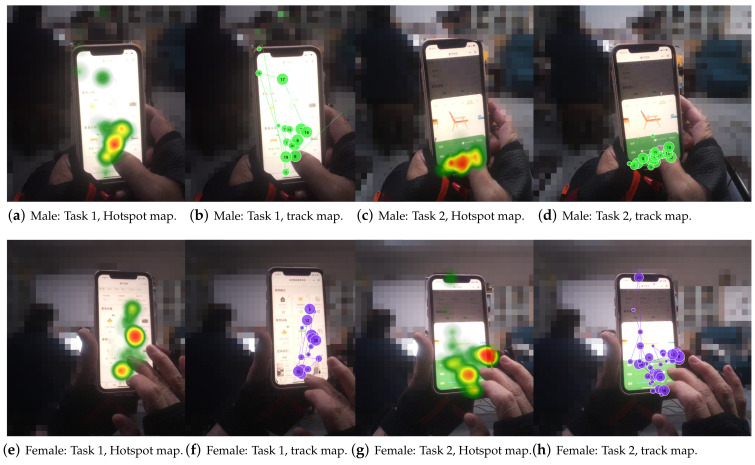
Eye movement hotspot maps and eye movement trajectory maps in elderly subjects of different genders. The darker the colour on the heat map, the more attention the area received. The numbers in the trajectory diagram represent the order in which the eyes are gazing.

**Figure 5 ijerph-19-16937-f005:**
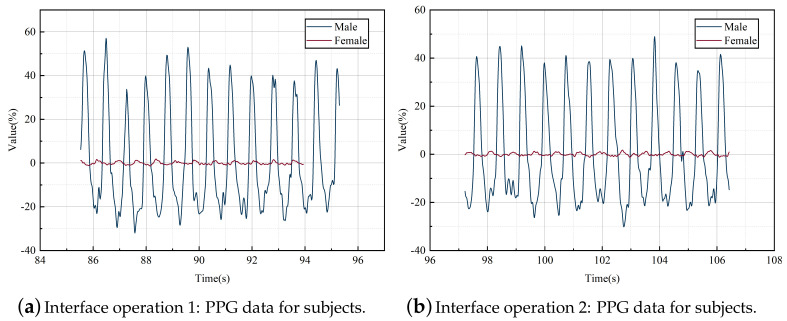
Changes in subject PPG data during interface operation. A graph of the changes in the PPG of subjects of different genders throughout the experiment, produced after processing by Origin data analysis software.

**Table 1 ijerph-19-16937-t001:** Subjects’ basic details.

Project	Female Subjects (n = 21)				Male Subjects (n = 21)			
	Age	Height (m)	Weight (kg)	BMI	Age	Height (m)	Weight (kg)	BMI
Average	64.36	1.59	64.92	25.54	66.25	1.71	72.43	24.62
SD	3.64	0.05	10.33	3.16	1.56	0.04	4.32	1.66
Max	71.00	1.68	81.50	31.83	69.00	1.81	79.00	27.33
Min	60.00	1.50	57.00	19.55	64.00	1.65	65.00	21.97

**Table 2 ijerph-19-16937-t002:** Validation interface 1: Frequency domain analysis of male subjects’ eye movement data. Note: 1: eye-velocity processing data, 2: left-pupil processing data, 3: right-pupil processing data, 4: mean-pupil processing data.

Experimenters	Projects	Max	Min	Average	SD	Variance	Mean	Median
Male	1	485.57	−1.00	25.60	61.01	3721.63	458.56	61.01
	2	5.16	2.99	3.82	0.55	0.30	2.56	2.99
	3	5.18	2.73	3.59	0.60	0.37	2.49	2.73
	4	4.97	2.88	3.71	0.57	0.32	2.49	2.88
Female	1	527.20	−1.00	15.83	43.47	1889.76	425.05	43.47
	2	3.53	2.67	3.26	0.12	0.01	1.92	2.67
	3	3.86	3.15	3.55	0.13	0.02	2.14	3.15
	4	3.86	3.15	3.55	0.13	0.02	2.14	3.15

**Table 3 ijerph-19-16937-t003:** Validation interface 2: Frequency domain analysis of male subjects’ eye movement data. Note: 1: eye-velocity processing data, 2: left-pupil processing data, 3: right-pupil processing data, 4: mean-pupil processing data.

Experimenters	Projects	Max	Min	Average	SD	Variance	Mean	Median
Male	1	410.06	−1.00	11.46	33.24	1105.15	311.78	33.24
	2	4.38	3.40	4.06	0.23	0.06	2.43	3.40
	3	4.07	3.15	3.78	0.22	0.05	2.25	3.15
	4	4.22	3.28	3.92	0.23	0.05	2.34	3.28
Female	1	442.09	−1.00	17.04	41.62	1732.10	432.91	41.62
	2	3.89	2.88	3.48	0.19	0.04	2.10	2.88
	3	4.50	3.21	3.84	0.24	0.06	2.37	3.21
	4	4.50	3.21	3.84	0.24	0.06	2.37	3.21

**Table 4 ijerph-19-16937-t004:** Variation in pupil distance between subjects of different genders during complex task interaction.

Subjects	Number of	Average Number	Saccades	Average Number	Time of
	Blinks (N)	of Blinks (N/s)	(N)	of Saccades(N/s)	Saccades(s)
Males	76	0.33	771	3.37	39.98
Females	73	0.36	692	3.42	28.52

**Table 5 ijerph-19-16937-t005:** Pupillary changes in subjects of different genders during complex task interaction. The pupil diameters of the different sexes were recorded during the experiment; we show the maximum, minimum, and mean values. The pupil diameter reflects the user’s eye fatigue and was therefore used as a measure.

Subjects	Average Pupil Diameter (mm)	Min Pupil Diameter (mm)	Max Pupil Diameter (mm)
Males	4.26	1.42	5.82
Females	3.79	2.57	5.38

**Table 6 ijerph-19-16937-t006:** HRV data for subjects of different genders in Task 1.

Tasks	Subjects	Mean IBI (ms)	Mean HR (bpm)	SDNN (ms)	RMSSD (ms)
Task 1	Males	793.75	76.00	31.08	41.34
	Females	740.63	81.00	46.12	75.30
Task 2	Males	772.73	78.00	29.12	51.35
	Females	752.76	80.00	67.94	111.58

## Data Availability

Not applicable.

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
