# Peer review of "Study of Ageing in Complex Interface Interaction Tasks: Based on Combined Eye-Movement and HRV Bioinformatic Feedback"

_ijerph, 2022, doi:10.3390/ijerph192416937_

Round 1

Reviewer 1 Report

1) I would suggest including a slightly more detailed description of the task and menus that participants were engaging with. I think the illustrations are really great, but a bit more detail on what they saw, what the purpose of the task was, etc, would help to better make sense of the study and what happened during the task. 

2) P. 4 " Eye velocity is generally related to attentional focus, and it is normally assumed that the higher the eye movement velocity, the less focused the attention."-  I think this statement needs some references. I am personally not familiar with how eye velocity can be used as a measure of attentional focus. Eye velocity is programmed based on the distance to the next fixation (velocity increases with the distance to the fixation). So, I think you need to elaborate more on this and provide references to support your argument. 

3) P.4 "During interface 1, the mean eye movement velocity of males was more significant than that of females, and males were more distracted. A variance test showed that the male eye-movement velocity column ancestor data was more discrete and 124 had higher Median values than the female." - Similar to what I mentioned above, I'm still not sure how this demonstrates that one group was more distracted. There is some research showing that peak velocity is related to arousal (Di Stasi, L. L., Catena, A., Canas, J. J., Macknik, S. L., & Martinez-Conde, S. (2013). Saccadic velocity as an arousal index in naturalistic tasks. Neuroscience & Biobehavioral Reviews37(5), 968-975), but I'm not aware of how such measures can show "distraction". I think the better way to approach this is to look at the fixations made on the screen and where they occurred. If the fixations were closer to the menu item participants were supposed to be engaging with, it would be reasonable to argue they were more engaged and less "distracted". Likewise, measures such as "dwell time" (sum of all fixation durations on an area of interest), would be quite telling in terms of where attention was located on the screen during the task. The heatmaps are nice as illustrations, but they have very limited value in terms of statistical analysis of the data and making sense of numerical patterns. 

4) The statistical results of the variance tests should also be reported in the text (at the moment, there is only a verbal description of the results). My understanding is that variance tests also test for differences in the distribution of the data, so doesn't this actually show that the data for one group is more variable than the other? At any rate, I would recommend reporting the statistical results more fully to make it easier to interpret your findings. 

5) I find it a bit hard to make sense of the numbers in Tables 2-4 because they show a lot of raw eye-tracking variables. But, without knowing more details about the task (e.g., where were participants looking, what do the coordinates represent?), it is difficult to make sense of them. My advice is to try to generate a bit more meaningful variables, such as fixation and saccadic measures that occurred on the screen as participants were engaged with the task. This can actually tell you what participants were looking at when they were engaging with the menus. 

6) You should report statistical test results for the analysis of pupil data on P.6 (the interpretation of the means is not enough to claim a difference).  

7) Discussion: It wasn't very clear how some of the conclusions are supported by the reported analyses. I think you should try to relate your conclusions more closely to the analyses that were carried out and be mindful of their limitations.

Thank you for the opportunity to read your interesting paper. 

Author Response

1) I would suggest including a slightly more detailed description of the task and menus that participants were engaging with. I think the illustrations are really great, but a bit more detail on what they saw, what the purpose of the task was, etc, would help to better make sense of the study and what happened during the task.

Answer: 

1) The experimenter in the field wore a wireless physiological monitoring device, which was used to monitor the HRV (Heart Rate Variablity) and EDA (Electrodermal Activity) data of the elderly subjects and determined their stress level during the task.

(2) The geriatric subject was fitted with Tobii Pro Glasses 2. Calibration of the oculomotor was performed prior to the start of the experiment. The subject was instructed to place the calibration card arm's length away, and the eye-tracking technique was used to calibrate the eye-tracking device. During the experiment, the subject operates the experimental handpiece installed with the program to adjust the experimental seat autonomously. During the mobile phone's operation, the subject must remain approximately 40 cm from the mobile phone's screen.

3) The elderly subject needs to sit in the experimental chair, find the button for seat adjustment on interface one (as shown in Figure 3a), find the corresponding icon and click on the adjustment slider.

4) After the adjustment slider was invoked, the elderly subject needed to adjust the second slider bar (shown in Figure 3b) and drag it to 125°.

5) At the end of the task, the field testers remove the experimental equipment for the elderly subject.

2) P. 4 " Eye velocity is generally related to attentional focus, and it is normally assumed that the higher the eye movement velocity, the less focused the attention."-  I think this statement needs some references. I am personally not familiar with how eye velocity can be used as a measure of attentional focus. Eye velocity is programmed based on the distance to the next fixation (velocity increases with the distance to the fixation). So, I think you need to elaborate more on this and provide references to support your argument.

Answer: 

To testify ‘Eye velocity is generally related to attentional focus, and it is normally assumed that the higher the eye movement velocity, the less focused the attention’, We have added relevant literature to support.

Campenhausen found that individuals who focused on a single stimulus experienced a significant increase in eye velocity and a decrease in look nystagmus. Studies have shown that eye movement indices such as peak sweep speed and blink frequency are related to the attentional control process. To further investigate the specific grading of this process, Bachurina experimentally investigated the relationship between eye movements and mental attention tasks, concluding that peak sweep speed and blink frequency decreased with mental attention demands and were negatively correlated with self-evaluation of mental work. Eye movement velocity is also used in practical occupational safety assessments to assess the attentional focus of actual workers. Zhu used video capture of workers working on safe and unsafe behaviours to determine the level of attention by eye movements. It was found that workers' attention levels were higher during high-risk operations than in safe operations and that workers' eye movement speed was higher than in safe operations.

3) P.4 "During interface 1, the mean eye movement velocity of males was more significant than that of females, and males were more distracted. A variance test showed that the male eye-movement velocity column ancestor data was more discrete and 124 had higher Median values than the female." - Similar to what I mentioned above, I'm still not sure how this demonstrates that one group was more distracted. There is some research showing that peak velocity is related to arousal (Di Stasi, L. L., Catena, A., Canas, J. J., Macknik, S. L., & Martinez-Conde, S. (2013). Saccadic velocity as an arousal index in naturalistic tasks. Neuroscience & Biobehavioral Reviews, 37(5), 968-975), but I'm not aware of how such measures can show "distraction". I think the better way to approach this is to look at the fixations made on the screen and where they occurred. If the fixations were closer to the menu item participants were supposed to be engaging with, it would be reasonable to argue they were more engaged and less "distracted". Likewise, measures such as "dwell time" (sum of all fixation durations on an area of interest), would be quite telling in terms of where attention was located on the screen during the task. The heatmaps are nice as illustrations, but they have very limited value in terms of statistical analysis of the data and making sense of numerical patterns.

4) The statistical results of the variance tests should also be reported in the text (at the moment, there is only a verbal description of the results). My understanding is that variance tests also test for differences in the distribution of the data, so doesn't this actually show that the data for one group is more variable than the other? At any rate, I would recommend reporting the statistical results more fully to make it easier to interpret your findings.

Answer: (Uniform answers and amendments to points 3-4)

Based on the suggestions, we have filtered and eliminated data from the original charts 2 to 4. The suggestion is significant, so we have added a description of the data analysis method, shown below.

The eye movement data under each task module was subdivided into data for the left and right eye, comprising 4 sets of items. 1: eye velocity processing data, 2: left pupil processing data, 3: right pupil processing data, 4: mean pupil processing data. The data were mainly analysed for Max, Min, Average, SD, variance, Mean and Median. During the overall operation of interface 1, the primary interaction task was to find the intelligent sofa operation panel, which can be categorised as a complex task of finding and locating.

The main interaction task in Task 1 was to find the adjustment icon of the intelligent sofa, which can be considered a finding task in a complex panel. Compared to the eye velocity processing data during this task, it can be seen from Table 2 that the average for the male elderly subjects in completing this task was 25.6 compared to 15.83 for the female elderly subjects. The multiple data values suggest that males maintained high levels of concentration for extended periods than females when completing the finding task. The main interaction task in Task 2 was adjusting the tilt angle of the intelligent sofa, which can be categorised as a complex task of goal attainment. Compared to the eye velocity processing data during this task, it can be seen from Table 3 that the average of male elderly subjects in completing this task was 11.46 compared to 17.04 for female elderly subjects. The multiple data values suggest that females were more attentive than males in completing the task.

5) I find it a bit hard to make sense of the numbers in Tables 2-4 because they show a lot of raw eye-tracking variables. But, without knowing more details about the task (e.g., where were participants looking, what do the coordinates represent?), it is difficult to make sense of them. My advice is to try to generate a bit more meaningful variables, such as fixation and saccadic measures that occurred on the screen as participants were engaged with the task. This can actually tell you what participants were looking at when they were engaging with the menus.

6) You should report statistical test results for the analysis of pupil data on P.6 (the interpretation of the means is not enough to claim a difference).

Answer: (Uniform answers and amendments to points 5-6)

The original Tables 2 to 4 contained 20 variables in eye movement data, but our analysis of the 20 variables needed to be more thorough. Therefore, the data was filtered, and four eye movement data were retained to support the final output.

We also added the blinks of subjects of different genders while performing the experimental manipulations.

Blink behaviour was often used to indicate internal and external attentional focus. Blink interrupts visual input, temporarily inhibits visual processing, and was reduced when the task requires processing external visual information. Wide eyes were associated with the type of experimental task while browsing tasks were external tasks requiring frequent sweeps. Annerer, in order to explore the relationship between wide eyes and attention in different task situations, used experiments to simulate different task types and verified through machine learning that in visuospatial tasks, the external focus of attention was associated with more wide eyes than internal attentional focus . Furthermore, the sweeping gaze remained consistently high in high-attentional situations. This experiment recorded the number of blinks, the average number of blinks, saccades, the average number of saccades, and the time of saccades for different gender subjects while performing complex external interface operations, as shown in Table 4. The average number of saccades for male elderly subjects was 3.37, while for female elderly subjects, it was 3.42. Therefore, this indicates that female elderly subjects were more attentive than male elderly subjects throughout the experiment.

7) Discussion: It wasn't very clear how some of the conclusions are supported by the reported analyses. I think you should try to relate your conclusions more closely to the analyses that were carried out and be mindful of their limitations.

Answer: 

The original conclusion section we have divided it into three sections: experimental conclusions, experimental design insights and experimental shortcomings and perspectives.

In experiments, it has been observed that older users often experience negative emotions, such as nervousness, when interacting with complex tasks. Therefore, the interface design should be simple and display only minor task information and interface elements. In this experiment, real-time data collected from multiple physiological signals (eye movement data, HRV data, EDA data) was used to judge the following conclusions: 1) Older users look at icons first before focusing on text parts during human-computer interaction. 2) When searching for complex tasks, female subjects first navigate through the global page before looking for basic tasks. 3) Females' reading habits are top-down. Male subjects were used to looking directly for the critical task rather than focusing excessively on the global interface. 4) Males were more focused than females when performing a finding task with a complex interface and remained focused throughout. 5) During interaction with complex tasks, female subjects experienced more minor mood swings and tension than male subjects

Based on the above studies, simple icons are preferred over complex verbal annotations when adapting to complex interface designs. In complex task interactions, the operational steps will be further simplified to reduce the operational difficulty and eliminate the uneasiness of older users. When personalising for male and female older users, the male older user interface should be more task-finding, with a simplified and eye-catching command panel. When personalising the user interface for female users, the user interface should be oriented towards operational tasks, with the Find task module being visible and easy to find. This will optimise the experience of older users and improve their performance.

There are still things that could be improved in this experiment. 1) The subjects operated the mobile phone independently during the experiment. To a certain extent, their ability to operate electronic devices may slightly impact the results. 2) The distance between the subject's eyes and the screen was controlled at 40cm. However, in the experiment, the subject's habits of using electronic products, such as leaning forward and leaning back, will be studied in depth based on the pupil's vertical distance and other data. 3) This experiment lacked personal scale collection for the subjects, and the subsequent experiment will combine subjective and objective to establish a multi-dimensional assessment system. 4) This experiment combines three channels of eye movement, HRV, and EDA signals to investigate the behavioural characteristics of elderly male and female subjects of different genders when manipulating complex tasks. A follow-up experiment will further integrate EEG (Electroencephalogram) and other psychological load indicators to investigate older subjects' performance characteristics in complex tasks.

Reviewer 2 Report

An interesting research but the many of the conclusions are not obvious from the presented data and the paper needs additional proofreading.  More specifically below:

1) The meaning of some expressions or sentences is not clear. For example:

--Line 80-81: Large print loud?

--Line 104: What is averaged subject?  probably average pupil diameter not subject?

-- Line 113: EDA is not explained

-- Line 116: unfinished sentence? By counting the data of each type... as shown in Table 2-5.  What is achieved/shown/...by counting the data?

2) Data in Tables 2-5 is not clearly explained (what is left pupil processing data, etc.?)  Furthermore, what are validation interface 1 and 2? Probably the ones from Figure 3, but in the Figure 3 they are named differently?

3) Unfinished sentence  starting in Line 140? (In the search task for interface 1, as shown in Figure 4.)

4) Figure 4 does not show any male or female data, but in Line 141 there are some statements related to male and female patterns of behaviour

5) The conclusions in the Discussion are not supported by the presented data, especially the ones related to the focus of male and female subjects in the user interface.

6) Table 1 shows 21 male subjects, while in the text there are only 18 male subjects?

Author Response

1) The meaning of some expressions or sentences is not clear. For example:

--Line 80-81: Large print loud?

--Line 104: What is averaged subject?  probably average pupil diameter not subject?

-- Line 113: EDA is not explained

-- Line 116: unfinished sentence? By counting the data of each type... as shown in Table 2-5.  What is achieved/shown/...by counting the data?

Answer:

We are grateful to the experts for suggesting any sloppy areas in the article, for which we have revised them all.

--Line 80-81: Smartphone: full screen, 6.22 in., FHD+1, 520 720 px.

--Line 104: The average pupil diameter of the  subjects was similar in that it increased during the first part of the experiment, peaked, and then began to decrease.

-- Line 113: EDA (Electrodermal Activity)

-- Line 116: This section of the original text has been extensively revised and can be consulted: By counting the data of each type of eye movement of the subjects of different genders while performing the interface operations, as shown in the Table 2-3. Table 2 represents eye movement data for subjects of different genders during manipulation task 1, and Table 3 represents eye movement data for subjects of different genders during manipulation task 2.

2) Data in Tables 2-5 is not clearly explained (what is left pupil processing data, etc.?)  Furthermore, what are validation interface 1 and 2? Probably the ones from Figure 3, but in the Figure 3 they are named differently?

Answer:

The original Tables 2 to 4 contained 20 variables in eye movement data, but our analysis of the 20 variables needed to be more thorough. Therefore, the data was filtered, and four eye movement data were retained to support the final output.

We also added the blinks of subjects of different genders while performing the experimental manipulations.

Blink behaviour was often used to indicate internal and external attentional focus. Blink interrupts visual input, temporarily inhibits visual processing, and was reduced when the task requires processing external visual information. Wide eyes were associated with the type of experimental task while browsing tasks were external tasks requiring frequent sweeps. Annerer, in order to explore the relationship between wide eyes and attention in different task situations, used experiments to simulate different task types and verified through machine learning that in visuospatial tasks, the external focus of attention was associated with more wide eyes than internal attentional focus . Furthermore, the sweeping gaze remained consistently high in high-attentional situations. This experiment recorded the number of blinks, the average number of blinks, saccades, the average number of saccades, and the time of saccades for different gender subjects while performing complex external interface operations, as shown in Table 4. The average number of saccades for male elderly subjects was 3.37, while for female elderly subjects, it was 3.42. Therefore, this indicates that female elderly subjects were more attentive than male elderly subjects throughout the experiment.

3) Unfinished sentence  starting in Line 140? (In the search task for interface 1, as shown in Figure 4.)

Answer: 

This is not an unfinished sentence, but we have adapted and added it to this paragraph for the reader to better understand it.

In the search task for interface 1, the hotspot map shows that female subjects were more likely to be attracted to the rest of the interface, combined with the trajectory map.

4) Figure 4 does not show any male or female data, but in Line 141 there are some statements related to male and female patterns of behaviour

Answer:

This was an oversight on our part, as we needed to make it clear in the text what these images represented, thus leading to some misunderstanding on the reader's part. Figures 4a-d show the hotspot and eye-tracking graphs for men during page browsing and task completion, while Figures 4e-h show the hotspot and eye-tracking graphs for women during page browsing and task completion.

The name of Figure 4 has been changed accordingly in the hope that it will better help the reader to understand its actual meaning.

5) The conclusions in the Discussion are not supported by the presented data, especially the ones related to the focus of male and female subjects in the user interface.

Answer:

The original conclusion section we have divided it into three sections: experimental conclusions, experimental design insights and experimental shortcomings and perspectives.

In experiments, it has been observed that older users often experience negative emotions, such as nervousness, when interacting with complex tasks. Therefore, the interface design should be simple and display only minor task information and interface elements. In this experiment, real-time data collected from multiple physiological signals (eye movement data, HRV data, EDA data) was used to judge the following conclusions: 1) Older users look at icons first before focusing on text parts during human-computer interaction. 2) When searching for complex tasks, female subjects first navigate through the global page before looking for basic tasks. 3) Females' reading habits are top-down. Male subjects were used to looking directly for the critical task rather than focusing excessively on the global interface. 4) Males were more focused than females when performing a finding task with a complex interface and remained focused throughout. 5) During interaction with complex tasks, female subjects experienced more minor mood swings and tension than male subjects

Based on the above studies, simple icons are preferred over complex verbal annotations when adapting to complex interface designs. In complex task interactions, the operational steps will be further simplified to reduce the operational difficulty and eliminate the uneasiness of older users. When personalising for male and female older users, the male older user interface should be more task-finding, with a simplified and eye-catching command panel. When personalising the user interface for female users, the user interface should be oriented towards operational tasks, with the Find task module being visible and easy to find. This will optimise the experience of older users and improve their performance.

There are still things that could be improved in this experiment. 1) The subjects operated the mobile phone independently during the experiment. To a certain extent, their ability to operate electronic devices may slightly impact the results. 2) The distance between the subject's eyes and the screen was controlled at 40cm. However, in the experiment, the subject's habits of using electronic products, such as leaning forward and leaning back, will be studied in depth based on the pupil's vertical distance and other data. 3) This experiment lacked personal scale collection for the subjects, and the subsequent experiment will combine subjective and objective to establish a multi-dimensional assessment system. 4) This experiment combines three channels of eye movement, HRV, and EDA signals to investigate the behavioural characteristics of elderly male and female subjects of different genders when manipulating complex tasks. A follow-up experiment will further integrate EEG (Electroencephalogram) and other psychological load indicators to investigate older subjects' performance characteristics in complex tasks.

6) Table 1 shows 21 male subjects, while in the text there are only 18 male subjects?

Answer:

Many thanks to the experts for pointing out our mistake, which was a clerical error.

21 males and 21 females in their 60s to 75s, all right-handed, with normal or corrected vision, were chosen for this study, as shown in Table 1. It is worth stating that all subjects were required to be free of eye disease and in good visual condition. For ethical compliance, all subjects who participated in the research signed an informed consent form, and all experiments were carried out with the subjects' consent.

Round 2

Reviewer 2 Report

The paper has been improved, I would just recommend another proof reading and corrections of the English language by an English native speaker or similar.